# Intestinal Tissue, Digestive Enzyme, and Antioxidant Enzyme Activities in the Early Development Stage of Endangered *Brachymystax tsinlingensis*

**DOI:** 10.3390/ani14203042

**Published:** 2024-10-21

**Authors:** Rongqun Song, Zhenlu Wang, Shaoqing Lin, Xingchen Guo, Yizhou Wang, Lin Zhang, Huan Ye, Jian Shao

**Affiliations:** 1Laboratory of Fishery Resources and Environmental Protection, College of Animal Science, Guizhou University, Guiyang 550025, China; songrongqun2021@163.com (R.S.);; 2The Key Laboratory of Animal Genetics, Breeding and Reproduction in the Plateau Mountainous Region, Ministry of Education, College of Animal Science, Guizhou University, Guiyang 550025, China; 3Special Fisheries Research Institute, Guizhou University, Guiyang 550025, China; 4Tibet Animal Husbandry Service Center, Lhasa 850000, China; 5Key Laboratory of Freshwater Biodiversity Conservation, Ministry of Agriculture and Rural Affairs, Yangtze River Fisheries Research Institute, Chinese Academy of Fishery Sciences, Wuhan 430223, China

**Keywords:** *Brachymystax tsinlingensis*, early development, intestinal tissue structure, digestive enzymes, antioxidant enzymes

## Abstract

The wild population of *Brachymystax tsinlingensis* is a Grade II protected species in China. This population faces slow growth, harsh habitats, serious individual miniaturization, and a low survival rate of fry during its early developmental stage. To improve the aquaculture efficiency of this species, and thus increase the survival rate, protect species resources, and enrich species numbers, the present study was conducted to investigate the changes in intestinal tissue structure, digestive enzymes, malondialdehyde content, and antioxidant enzyme activities and their patterns during the early developmental stage of *Brachymystax tsinlingensis*. This information can be applied to increase the survival rate of fry, protect the species resources, and enrich the species’ quantity.

## 1. Introduction

The growth and development of early embryos or juveniles affect the subsequent growth of fish, and the development of digestive system tissues is also different due to the three endogenous, mixed, and exogenous vegetative periods. The transition of fry from endogenous yolk sac nutrition to mixed nutrition and then to exogenous nutrition is often a period of high mortality. The primary challenge in artificial fry breeding is lowering the mortality rate. To address this, we must study the digestive system development of fry and ensure they can effectively digest and absorb nutrients from their food [1]. Further, it is important to consider the early stages of life in the study of ontogeny to understand nutrient storage and circulation, as most changes occur during this time [2]. Information about the activity of intestinal enzymes (amylase, protein, trypsin, chymotrypsin, and lipase) during ontogeny is essential to understanding the digestive efficiency of species at specific developmental stages [3]. Additionally, digestive enzyme activity can be used as an indicator of larval food acceptance and, to some extent, serves as an indicator of the digestive capacity in relation to the type of feed offered [4]. Falk-Petersen [5] pointed out that studying information on endocrine and immunocompetent tissue and organ function during the early developmental stages of fish is an area that deserves to be focused on in the future. Salimian [6] used triploids to induce physiological and immune characteristics of Oncorhynchus mykiss during early developmental stages (fertilized eggs, eyed eggs, and fry) and showed that triploid induction does not compromise the immune system of these fish. Perrichon’s study of early individual development in *Ammodytes* marinus improved the quality of the juvenile stage in intensive aquaculture systems and also assessed the nutritional requirements of juvenile fish [7]. These studies suggest that studying gut organization, digestive enzymes, and antioxidant capacity during the early developmental stages of fish could be important for the captive breeding and rearing of endangered fish species. 

*Brachymystax tsinlingensis* Li, 1966, belonging to the order Salmoniformes, the family Salmonidae, and the genus *Brachymystax*, commonly known as five-color fish or plum fish, is endemic to China and has economic, ecological, and scientific value [8,9,10,11,12]. This species originated from the northern frigid zone of Eurasia and is a residual fish of marine creatures that moved southwards to China owing to glacial advances during the quaternary glaciation (Li, 1984). *Brachymystax tsinlingensis* has extremely specific requirements for its ecological environment and is only found in the cold-water streamlet of the Qinling Mountains, China [13]. In recent years, research has shown that Tsinling lenok trout have specific habitat water temperature requirements and are susceptible to the impacts of global climate change and human activities [14]. The population of *Brachymystax tsinlingensis* has decreased rapidly, accounting for only 10% of the original distribution, and its living altitude has increased by 200–300 m [15]. *Brachymystax tsinlingensis* is a carnivorous fish, feeding on shrimp, aquatic insects, and aquatic invertebrates as well as small fish and terrestrial insects blown down by the wind; feeding mostly occurs in the morning when the light is weak and in the evening when there are more flying insects [8,9,16,17,18,19]. Unfortunately, the wild resources of *Brachymystax tsinlingensis* have declined dramatically in recent years, and it has been listed as “vulnerable” in the China Red Data Book of Endangered Animals and classified as a second-class state-protected wild animal in China [16,20,21,22]. In recent years, a large number of scholars have conducted research on *Brachymystax tsinlingensis*, including its breeding [23], genetic distribution [24,25,26,27], underlying physiological characteristics [28,29,30], and external influences on stress [31,32]. However, little is known about the activities of intestinal tissue, digestive enzymes (total protein, lipase, trypsin, and amylase), and antioxidant enzymes (catalase, glutathione peroxidase, and superoxide dismutase) in the early development stage of *Brachymystax tsinlingensis*.

In this study, the intestinal tissues and the activities of digestive enzymes, malondialdehyde content, and antioxidant enzymes in the early developmental stages of *Brachymystax tsinlingensis* were assessed through histological and biochemical analyses. To assess digestive function, we measured the total protein concentration of the participating gut and the activity of several key digestive enzymes (lipase, trypsin, and amylase) as well as the gut malondialdehyde content and antioxidant indicators of the activity of several key antioxidant enzymes (catalase, glutathione peroxidase, and superoxide dismutase). These results could provide scientific guidance for the artificial seed cultivation technology of *Brachymystax tsinlingensis* and help us enrich the population of *Brachymystax tsinlingensis*.

## 2. Materials and Methods

### 2.1. Laboratory Animals

Experiments were conducted from April to July 2023 at the *B. tsinlingensis* Artificial Breeding Experimental Base in the Qinling Mountain region (Shaanxi Province, China). The experimental samples were all from the same batch of parent *B. tsinlingensis* fertilized eggs, which were cultivated in 3.0 m × 0.4 m × 0.25 m seedling tanks, in which the water temperature was 13.66 ± 1.89 °C, pH was 7.08 ± 0.16, and dissolved oxygen content was 7.56 ± 0.83 mg/L.

### 2.2. Specimen Collection

According to the changes in intestinal physiology and biology in the early development stage of *B. tsinlingensis*, ten sampling points were established over three stages: endogenous nutrition (7, 10, and 11 days after hatching), mixed nutrition (13 and 19 DAH), and exogenous nutrition (31, 33, 39, 45, and 73 DAH). From 13 to 31 DAH, the animals were fed with harvest worms; at 19 DAH, yolk sacs disappeared; at 33–39 DAH, they were fed with water earthworms; and at 45–73 DAH, they were fed with feed. The endogenous nutrients were based on yolk sacs, and the main components were proteins. The mixed nutrients were based on yolk sacs and aquatic earthworms, and the main components were proteins and a variety of essential amino acids. Lastly, the exogenous nutrients were from an artificial feed (ALLER FUTURA EX GR (crude protein ≥ 64%, crude fat ≥ 8%, coarse fiber ≤ 3%, water content ≤ 8.5%, crude ash content ≤ 12%, calcium ≥ 0.8%, total phosphorus ≤ 2%, and amino acid ≥ 3.3%)), which is from Qingdao, China. Each sample was collected two hours after feeding, with three samples randomly collected and fixed in 4% paraformaldehyde, and the physiological development of intestinal tissues was observed. Six tails were collected and stored in liquid nitrogen, and the activities of intestinal digestive enzymes, malondialdehyde content, and antioxidant enzymes were subsequently detected.

### 2.3. Experimental Methods

#### 2.3.1. H. E.-Stained Sections

The fixed tissues were soaked in 4% paraformaldehyde, paraffin-embedded and sectioned (5–7 μm), deparaffinized by xylene, stained with hematoxylin–eosin, air-dried, and then covered and observed with an optical microscope (Nikon 80i) in Shanghai, China [33]. Intestinal villi height, villi width, and muscle thickness were measured using ImageJ (win64), and 10 points were randomly sampled for each sample and replicated three times.

#### 2.3.2. Biochemical Indicators Testing

Intestinal samples were accurately weighed to 0.1~0.2 g, added with pre-cooled saline at a ratio of weight (g) to volume (mL) = 1:10, processed by a high-speed grinder at 2500 rpm, and centrifuged for 10 min, and the supernatant was taken for determination. Total protease, trypsin, lipase, and amylase were selected as markers of digestive enzymes; malondialdehyde as a marker of oxidative stress; and superoxide dismutase, catalase, and glutathione peroxidase as markers of the antioxidant defense system. All kits were from Nanjing Jiancheng Company (Nanjing, China). The total protein concentration was detected by using the Thomas Brilliant Blue method (A045-2-1), the trypsin activity by the ultraviolet colorimetry colorimetric method (A080-2), the lipase activity by the colorimetric method (A054-1-1), the amylase activity by the amylose-iodine colorimetric method (C016-1-1), and the malondialdehyde content by the thiobarbituric acid method (A003-1-1) of Nanjing Jiancheng Biotechnology’s kits. Catalase activity was detected by the colorimetric method (A084-1-1), glutathione peroxidase activity was detected by the colorimetric method (A005-1-2), and superoxide dismutase activity was detected by the water-soluble tetrazolium salt method (A001-3-2). The absorbance was measured using a 752 N ultraviolet colorimetry spectrophotometer (Shanghai Yidian Analytical Instruments Co., Ltd., Shanghai, China). Finally, the relevant activities were calculated using a formula. The experimental procedures were carried out according to the manufacturer’s instructions and were repeated six times to ensure consistency [34,35].

### 2.4. Data Processing and Analysis

The experimental results are expressed as mean ± standard deviation (mean ± SD). One-way ANOVA and nonparametric tests were performed using SPSS 17.0 software, and Duncan’s multiple comparison test and the Kruskal–Wallis test were used to analyze the significance of differences between groups. Statistically significant differences were found when *p* < 0.05.

## 3. Results

### 3.1. Microscopic Observation of Intestinal Tissue Structure in the Early Development Stage of Brachymystax tsinlingensis

Microscopic observation showed that the intestinal tissue structure of *B. tsinlingensis* had four layers, namely a mucosal layer, submucosa, muscle layer, and serous membrane, at 7 DAH, which were mucosal folds. The number of goblet cells gradually increased with growth and development, and at 73 DAH, there was an intestinal gland structure (Figure 1).

The microstructure of the intestinal tissue was measured and analyzed (Table 1). The height of the intestinal villi had a maximum value at 45 DAH (187.31 ± 8.89 μm), and the value at 11 DAH was significantly lower than that at 45 DAH (*p* < 0.05). Simultaneously, the maximum value of intestinal villi width was found at 45 DAH (69.33 ± 2.42 μm), with a significant difference between the values at 13 DAH and 45 DAH (*p* < 0.05). The thickness of the intestinal myometrium had a maximum value at 73 DAH (30.54 ± 3.63 μm), which was significantly higher than that at 11 DAH (*p* < 0.05).

### 3.2. Changes in Intestinal Digestive Enzymes in the Early Development Stage of Brachymystax tsinlingensis

The results demonstrated that the changes in intestinal digestive enzymes in the early development stage of *B. tsinlingensis* showed an upward trend. The total protein concentration showed an overall upward trend, even after the yolk sac disappeared at the stage of feeding earthworms (19–31 DAH) and at the stage of switching to feed (39–45 DAH). More importantly, there were significant differences between 7 DAH and 11, 13, 19, 31, 33, 39, and 45 DAH (*p* < 0.05), as the values at 39, 45, and 73 DAH were significantly higher than that at 11 DAH (*p* < 0.05). LPS activity increased after the mixed trophic period, which decreased after reaching a peak at 39 DAH. Moreover, there was no significant difference between the various stages (*p* > 0.05). TPS activity showed an upward trend during the exogenous vegetative period and decreased after reaching a peak at 39 DAH. The value at 73 DAH was significantly lower than those at 33, 39, and 45 DAH (*p* < 0.05). AMS activity showed an upward trend during the exogenous vegetative period and decreased after reaching a peak at 45 DAH. The value at 73 DAH was significantly lower than those at 33, 39, and 45 DAH (*p* < 0.05), and that at 45 DAH was significantly higher than the value at 7 DAH (*p* < 0.05). The proportion of AMS increased enormously from 33 DAH to 45 DAH in the early development stage of *B. tsinlingensis*, but the relative percentage of TRS was relatively stable in the overall amounts of intestinal digestive enzymes during the experiment (Figure 2 and Figure 3).

### 3.3. Changes in Intestinal Malondialdehyde Content and Antioxidant Enzymes in the Early Development Stage of Brachymystax tsinlingensis

The results showed that the changes in intestinal antioxidant enzymes in the early development stage of *B. tsinlingensis* were as follows: the content of malondialdehyde showed an overall upward trend, increasing from the disappearance of the yolk sac in the exogenous vegetative period to the stage of feeding earthworms (19–39 DAH) and the feed stage (45–73 DAH); the level at 73 DAH was significantly higher than that at 7, 19, 31, and 39 DAH (*p* < 0.05). CAT activity decreased after the disappearance of the yolk sac (19 DAH) but increased after reaching the lowest point at 39 DAH, and there was no significant difference in the other groups (*p* > 0.05). The activity of GSH-Px showed a decreasing trend overall. Specifically, there was an upward trend from the disappearance of the yolk sac to the stage of feeding earthworms (19–33 DAH) and the stage of switching to feed (39–73 DAH) after the disappearance of the yolk sac in the exogenous trophic period, with no significant difference among the phases (*p* > 0.05). The activity of SOD showed a decreasing trend on the whole. The enzyme activity was slow from the disappearance of the yolk sac to the earthworm feeding stage (19–33 DAH) in the exogenous trophic period. However, the enzyme activity showed an upward trend in the final feed stage (45–73 DAH). There was no significant difference between the groups in the mixed trophic period (13–19 DAH) and the exogenous trophic period (31–73 DAH); the activities at 39 and 45 DAH were significantly lower than those at 7 and 10 DAH (*p* < 0.05). The relative percentages of the four intestinal antioxidant enzymes were all relatively stable in the overall levels of intestinal digestive enzymes, even at different vegetative periods (Figure 4 and Figure 5).

## 4. Discussion

The intestine is an important place in which food is constantly digested, absorbed, and transported backwards, with the residue finally excreted as feces [36,37]. Juvenile *B. tsinlingensis* were found to have a four-layer structure including a mucosal layer, submucosa, muscle layer, and adventitia at 7 DAH, with fold formation to increase the absorption area, as well as goblet cells in the mucosal layer that gradually increased with development and growth [19,29,38,39]. Surprisingly, intestinal gland tissue was spotted at 73 DAH, which is inconsistent with previous research reports of the digestive system of *B. tsinlingensis* [19,29]. Intestinal morphology, such as villi height or epithelial thickness, is an indicator of the intestinal health of animals, including fish [40]. In this study, the height, width, and thickness of intestinal villi were significantly different when the yolk sac began to absorb and disappeared. Some studies have found that the increase in the number of goblet cells will increase the secretion of intestinal mucus in juveniles of largemouth bass (*Micropterus salmoides*) under ammonia nitrogen stress, leading to an increase in villi width and muscle thickness, which may be symptoms of oxidative stress and inflammation in the intestine [41]. A study found that feeding ferulic acid to genetically modified tilapia resulted in a decrease in villi height and width, possibly due to oxidized fish oil in the feed damaging the intestinal structure and reducing nutrient absorption [42]. In this study, the basic intestinal tissue structure of *B. tsinlingensis* was found at the endogenous trophic stage at 7 DAH, and the height, width, and thickness of the intestinal villi changed with the growth and development of the fry. This showed that the change in intestinal villi was related to not only the species itself but the condition of nutrient absorption mode and bait feeding. In conclusion, understanding the structural development of the gut is beneficial to *B. tsinlingensis*, guiding mass production and thus conserving the species’ resources.

The intestinal tract plays an important role in organs for nutrient absorption and digestive enzymes; importantly, measuring the activity of digestive enzymes in the intestine can reflect digestion in the body to a certain extent [43]. The development of digestive enzymes reflects the functional development and digestive capacity of the digestive tract of organisms. Therefore, they are often used as physiological biomarkers to assess the nutritional status of fish in the early stages of life development [44]. In this study, the total protein concentration of *B. tsinlingensis* showed an overall upward trend during early development. Consistent with a study on the change in digestive enzyme patterns in carp (*Cirrhinus mrigala*) and a study on the digestive enzyme development of juvenile perch (*Perca fluviatilis*), it was indicated that juveniles needed a large amount of protein for growth and development [45,46]. However, with the yolk sac disappearing, the total protein concentration rapidly decreased for more than 10 days; it may be the adjustment of bait adaptation of juveniles themselves during exogenous nutrition intervention. Meanwhile, the lipase activity began to increase after the mixed trophic period, indicating that the juveniles also absorbed the nutrients of the abundant insects when absorbing the nutrients of the yolk sac, and it decreased after reaching the highest value at 39 DAH. Some studies showed an overall upward trend in the ontogeny of digestive enzymes of *Ompok bimaculatus* larvae and the changes in enzyme activity of Nile tilapia (*Oreochromis niloticus*) larvae fed with different CP levels [44,47]. In the characteristics of the digestive enzyme pattern of carp (*Cirrhinus mrigala*), it was found that the change trend of LPS activity was consistent, and the LPS activity was affected by dietary changes [44]. In our study, TPS activity showed a downward trend at the outset, and the early change in AMS activity was flat, which is consistent with the changes in the TPA and AMS activities of greater amberjack developed under a circadian rhythm [48]. The activities of TPS and AMS showed an upward trend during the exogenous vegetative period. The enzyme activity increased when the juveniles themselves had exhausted the nutrients in the yolk sac and switched to eating earthworms, which indicated that the digestibility of earthworm ingestion in juveniles increased. However, TPS activity decreased after reaching a peak at 39 DAH, and AMS activity decreased after reaching a peak at 45 DAH. The present study revealed that the conversion of endogenous nutrients to exogenous nutrients is a critical period in the early development stage of *B. tsinlingensis* as a carnivorous fish. Understanding the changes in digestive enzymes can support the timely supplementation of the nutrients required by the fry and promote their growth and development.

Oxidative stress can cause damage to body cells and tissues [33], and MDA is an indicator of oxidative stress and antioxidants [49,50]. Moreover, GSH-Px, SOD, and CAT are essentially the first line of defense against antioxidants, and their efficacy and roles are essential for the entire antioxidant defense approach [51,52]. A decreased malondialdehyde content along with increased activities of catalase, superoxide dismutase, and glutathione peroxidase reduce tissue oxidative stress damage [47,53,54]. In our study, we found that the response of the content of MDA showed an overall upward trend, indicating that the nutrient intake of juveniles had an oxidative effect on the fry itself during growth and development, which continued to increase. CAT activity showed a downward trend after the disappearance of the yolk sac. Moreover, it was speculated that there were fewer free radicals, such as hydrogen peroxide, in the fry at this time, which increased after the lowest value at 39 DAH. The activities of GSH-Px and SOD showed a decreasing trend on the whole, indicating that the antioxidant response mechanism was not obvious in the whole early developmental stage. GSH-Px showed an upward trend from the stage when the yolk sac disappeared to the stage of feeding earthworms. The CAT, GSH-Px, and SOD levels showed an overall decreasing trend. Therefore, the oxidative stress damage of *B. tsinlingensis* gradually became serious. Furthermore, it is speculated that the fry themselves do not form antioxidant response abilities during development. The effects of hypoxia and reoxygenation on oxidative stress, histological structure, and apoptosis in a new hypoxia-tolerant variety of blunt snout bream (*Megalobrama amblycephala*) have been reported [50], as well as age and sex differences in antioxidant enzyme activities in brown trout (*Salmo trutta*) [51]. Failure to catalyze free radicals such as H_2_O_2_ in time results in oxidative stress damage. In our study, the feeding of bait also caused oxidative stress to the fish, who could not reduce the damage of oxidative stress and remove the toxic effects of cells in time. Consistent with the results of the changes in oxidative status in yellowfin seabream (*Acanthopagrus latus*) larvae during development [55], the juveniles had low antioxidant responses and oxidative damage in the early stage, and oxidative damage was greater with growth and development.

## 5. Conclusions

In summary, the intestinal tissue of *B. tsinlingensis* formed a relatively complete digestive structure at 7 DAH in the early development stage, and there was little change in the subsequent development. It preceded the formation of intestinal gland tissue at 73 DAH. During the early development stage of *B. tsinlingensis*, the digestive enzyme and antioxidant enzyme activity trends have unique characteristics that are different from other species. Namely, the activities of digestive enzymes and antioxidant enzymes changed with the species growth and development and the transfer of bait for a long time. *B. tsinlingensis* could not reduce oxidative stress damage, especially in the early development period. This suggests that the absorption and digestibility of fish fry are improved to better adapt to slow growth and the harsh living environment. The results showed that it is necessary to pay close attention to the change in nutrient absorption mode of fish fry before 19 DAH, and we suggest that timely feeding of exogenous bait (live food) is needed to solve the problem of the high lethality rate of fish fry during the vital feeding period (19–45 DAH) of *B. tsinlingensis*. It is hoped that appropriate bait feeding can reduce oxidative stress damage in the subsequent captive breeding of *B. tsinlingensis*.

## Figures and Tables

**Figure 1 animals-14-03042-f001:**
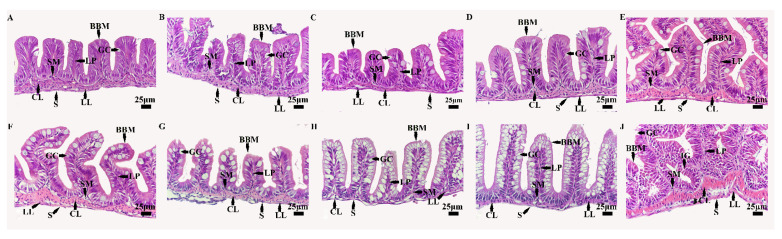
Intestinal tissue structure of *Brachymystax tsinlingensis*. (**A**) 7 days after hatching (DAH); (**B**) 10 DAH; (**C**) 11 DAH; (**D**) 13 DAH; (**E**) 19 DAH; (**F**) 31 DAH; (**G**) 33 DAH; (**H**) 39 DAH; (**I**) 45 DAH; (**J**) 73 DAH. S: serous; LL: longitudinal layer of smooth muscle; CL: circular layer of smooth muscle; SM: submucosa; LP: lamina propria; GC: goblet cells; BBM: brush border microvillus; IG: intestinal glands.

**Figure 2 animals-14-03042-f002:**
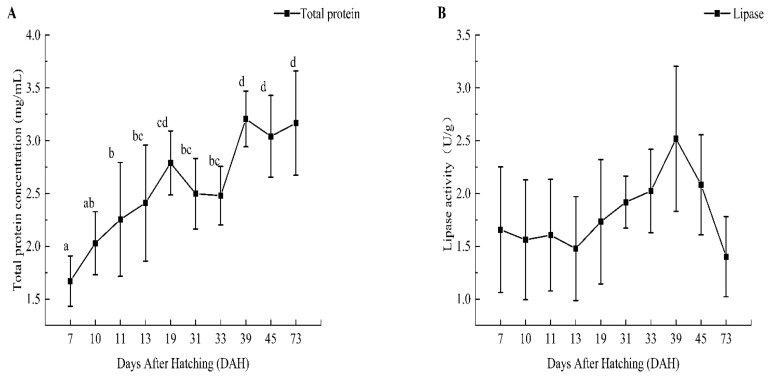
Intestinal digestive enzyme curves of total protein concentration (**A**), lipase activity (**B**), trypsin activity (**C**), and amylase activity (**D**). Lowercase letters indicate significant differences in digestive enzyme activity (*p* < 0.05).

**Figure 3 animals-14-03042-f003:**
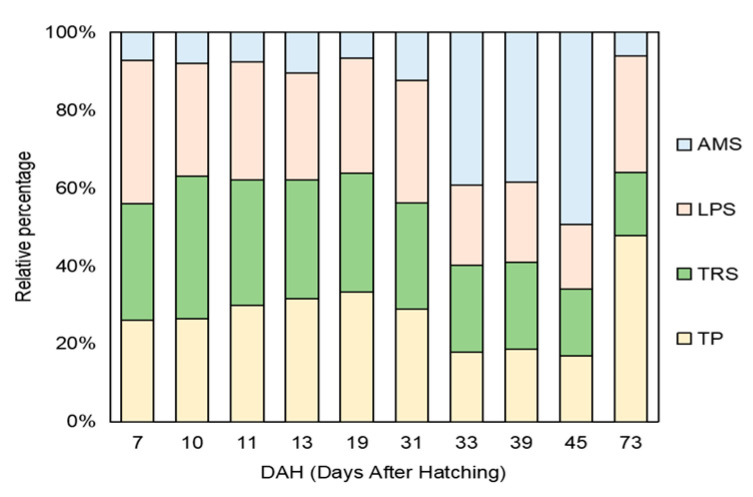
The relative percentage of total protein concentration, lipase activity, trypsin activity, and amylase activity in the early development stage of *Brachymystax tsinlingensis*.

**Figure 4 animals-14-03042-f004:**
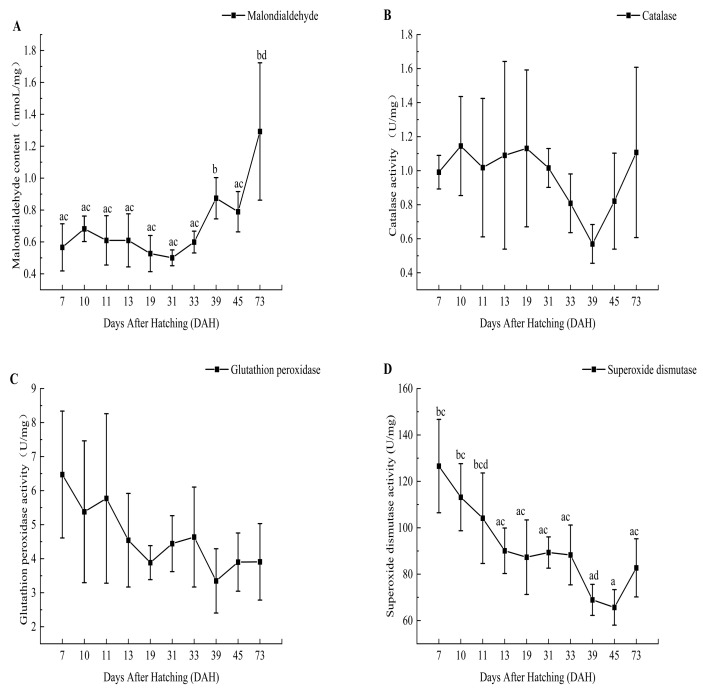
Malondialdehyde content (**A**) and antioxidant enzyme curve of catalase activity (**B**), glutathione peroxidase activity (**C**), and superoxide dismutase activity (**D**). Lowercase letters indicate significant differences in digestive enzyme activity (*p* < 0.05).

**Figure 5 animals-14-03042-f005:**
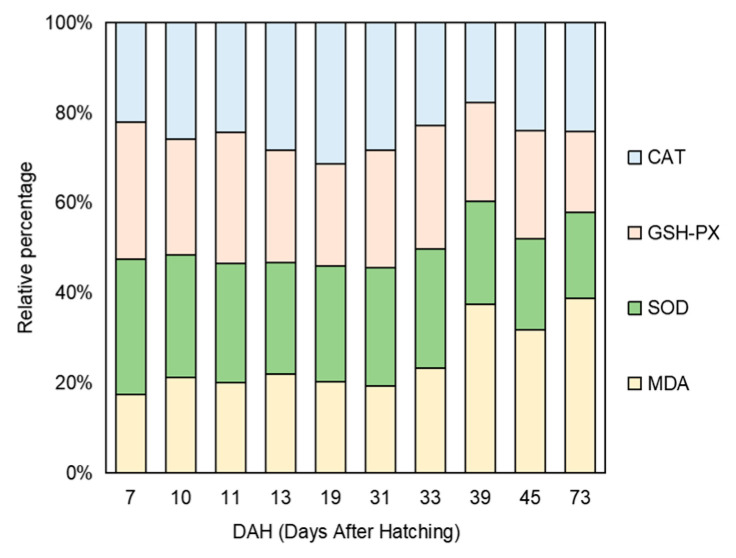
The relative percentage of malondialdehyde content, catalase activity, glutathione peroxidase activity, and superoxide dismutase activity in the early development stage of *Brachymystax tsinlingensis*.

**Table 1 animals-14-03042-t001:** Changes in intestinal tissue in the early developmental stage of *Brachymystax tsinlingensis*.

Developmental Stage		Items		
		Villus Height (μm)	Lint Width (μm)	Muscular Thickness (μm)
Endogenous nutrition	7	97.58 ± 13.26 ^ab^	53.30 ± 1.71 ^ab^	18.82 ± 2.03 ^ab^
	10	113.58 ± 19.55 ^ab^	58.08 ± 4.98 ^ab^	29.75 ± 0.38 ^ab^
	11	87.66 ± 7.09 ^a^	61.39 ± 8.05 ^ab^	15.33 ± 2.91 ^a^
Mixed nutrition	13	132.88 ± 15.05 ^ab^	52.44 ± 1.32 ^a^	20.06 ± 1.21 ^ab^
	19	154.94 ± 14.70 ^ab^	68.59 ± 6.50 ^ab^	22.95 ± 1.48 ^ab^
Exogenous nutrition	31	153.48 ± 41.42 ^ab^	60.10 ± 3.75 ^ab^	19.97 ± 3.09 ^ab^
	33	143.58 ± 10.86 ^ab^	57.32 ± 2.37 ^ab^	24.54 ± 2.96 ^ab^
	39	120.00 ± 6.73 ^ab^	55.72 ± 3.08 ^ab^	19.97 ± 1.94 ^ab^
	45	187.31 ± 8.89 ^b^	69.33 ± 2.42 ^b^	19.76 ± 5.84 ^ab^
	73	133.45 ± 16.17 ^ab^	63.97 ± 3.38 ^ab^	30.54 ± 3.63 ^b^

Different superscript letters in the same column indicate that the height, width, and thickness of intestinal villi at different stages are significantly different (*p* < 0.05).

## Data Availability

Data will be made available on request.

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
