# Peer review of "Intestinal Tissue, Digestive Enzyme, and Antioxidant Enzyme Activities in the Early Development Stage of Endangered Brachymystax tsinlingensis"

_animals, 2024, doi:10.3390/ani14203042_

Round 1
Reviewer 1 Report
Comments and Suggestions for Authors
The manuscript describes an interesting and valuable study concerning the intestine tissue in the early development stage of Endangered Brachymystax tsinlingensis. The motivation is interesting and some necessary work have been carried out by the authors. However, some deficiencies must be corrected before consideration for publication.
1. Add more ecological information about B. tsinlingensis and its role in the ecosystem.
2. The Introduction states that the purpose of the study is to explore the developmental characteristics of the B. tsinlingensis digestive system, which is crucial for artificial reproduction and population conservation. This part can be further refined, such as pointing out the specific problem or hypothesis to be solved.
3. Line 79: change “measure” to “measured”.
4. Line 169: change “Total protein” to “total protein”.
5. The results are not well-presented. The histological part is limited. in the conclusion Since the Discussion section includes mucus and goblet cells, it is suggested to add AB staining.
6. The Discussion section should discuss how oxidative stress affects the mechanism of B. tsinlingensis intestinal development.
7. Highlight the innovative points of this study in the Discussion section.
Comments on the Quality of English LanguageMinor editing of English language required
Author Response
Response to Reviewer 1 Comments
Point 1: Add more ecological information about B. tsinlingensis and its role in the ecosystem.
Response 1: As suggested, we have added more ecological information about B. tsinlingensis and its role in the ecosystem(Lines 73-83)
Point 2. The Introduction states that the purpose of the study is to explore the developmental characteristics of the B. tsinlingensis digestive system, which is crucial for artificial reproduction and population conservation. This part can be further refined, such as pointing out the specific problem or hypothesis to be solved.
Response 2: As suggested, we have identified specific issues to be addressed.
‘In order to explore the digestive system characteristics of Brachymystax tsinlingensis during early developmental stages and to solve the problem of high lethality of fry during the transgression period, which is crucial for the artificial propagation and population conservation of endangered fishes.’
Point 3. Line 79: change “measure” to “measured”.
Response 3: As suggested, we have changed “measure” to “measured”(Line 101).
Point 4. Line 169: change “Total protein” to “total protein”.
Response 4: As suggested, we have changed “Total protein” to “total protein”(Line 188).
Point 5. The results are not well-presented. The histological part is limited. in the conclusion Since the Discussion section includes mucus and goblet cells, it is suggested to add AB staining.
Response 5: We apologize for our negligence. We only made a developmental time series diagram of the intestinal tissue during bait switching, which simply describes the structure of the tissue, and did not perform AB staining on it.However, we will consider adding AB staining experiments in the subsequent work.
Point 6. The Discussion section should discuss how oxidative stress affects the mechanism of B. tsinlingensis intestinal development.
Response 6: Thank you for your suggestion. We can only describe whether tissues are damaged by oxidative stress in terms of changes in simple antioxidant enzyme activities, and we cannot discuss it in terms of specific mechanisms, but in subsequent studies we will design experiments from the perspective of mechanisms
Point 7. Highlight the innovative points of this study in the Discussion section.
Response 7: Thank you for pointing out those mistakes for us. We have already highlighted the innovations of this study in the discussion section.
Reviewer 2 Report
Comments and Suggestions for Authors
The manuscript entitled “Intestinal tissue, digestive enzyme and antioxidant enzyme activities in the early development stage of Endangered Brachymystax tsinlingensis” by Rongqun Song et al. reports the development characteristics of the digestive system, digestive enzymes and antioxidant enzyme activities in the early development stage of endangered fish such as Brachymystax tsinlingensis. The wild population of Brachymystax tsinlingensis is a Grade II protected animal in China. The species is characterised by slow growth, harsh habitat conditions, severe miniaturisation of individuals and low survival of juveniles in early developmental stages.
Basic reporting
In general, the experiments presented are well designed and contribute new data to the field. The choice of parameters used to characterise the developmental characteristics of the digestive and antioxidant systems seems appropriate for the study, although poorly described (and should be substantially revised). The quality of the presentation (the figures and a table) is satisfactory. The statistical analyses used were appropriate and sufficient for the purposes of this study. The main conclusions of this study are as follows: the changes in intestinal tissue, digestive enzymes and antioxidant enzyme activities 39 DAH of B. tsinlingensis are inseparable from different vegetative stages during early development, and these results of this study can provide a reference for the cultivation of B. tsinlingensis to restore the species resources. The conclusion is consistent with the data obtained. The text is clearly written and of interest to the reader. Some principal comments that need to be edited are detailed below. Overall, the work is suitable for publication in Animals.
Title
Line 3 – in ‘Endangered’ please use lower case
Abstract
Lines 18-22 – Obviously, the study of intestinal tissues, digestive and antioxidant enzyme activities and malondialdehyde content is intended to assess the well-being of early juveniles of B. tsinlingensis in order to improve the efficiency of aquaculture of the species, rather than to ‘increase survival’ or ‘protect species resources and enrich the species' quantity’. Please rephrase.
Line 31 – Please decipher the abbreviation TP.
Introduction.
Line 61 – Please provide full species name at the first mention as follows (indicating author) Brachymystax tsinlingensis Li, 1966.
Materials and Methods
Line 100 – Please provide a brief description of nutrient composition of the feed given to the fish at 45-73 DAH. Such data would facilitate discussion of the results obtained.
Line 102 – Was the fish collected whole (or as questionably stated 'six tails were collected') for enzyme activity assays? You refer to 'intestinal enzymes' throughout the text; please provide information on the separation of intestinal enzymes or the dissection of the intestinal contents.
Line 111 – Confusingly, you refer to Elisa tests for lipase, amylase, trypsin, catalase, glutathione peroxidase and superoxide dismutase, as you obviously measured the enzyme activities but not their contents. Please specify the method applied and give the names of the kits used.
Results
Line 143 – It is recommended that the authors consider dividing the data in Table 1 into developmental stages, which would include the following categories: endogenous nutrition (7, 10, 11 days after hatching), mixed nutrition (13, 19 DAH) and exogenous nutrition (31, 33, 39, 45, 73 DAH).
Line 149 – in ‘The total protease concentration showed an overall upward trend …’. Replace ‘protease’ with ‘protein’.
Lines 169, 202 – the figure captions ‘Trends in intestinal digestive enzymes of Total protein concentration’ and ‘Trends in intestinal antioxidant enzymes of Malondialdehyde content’ seem confusing and the word ‘trends’ is excessive; please rephrase.
Line 199 – Figure 4B (catalase) has an ‘a’ over all the points, which means no difference. You write: ‘CAT activity decreased after the disappearance of yolk sac (19 DAH), but that increased after reaching the lowest peak at 39 DAH, and there was no significant difference in other groups (p > 0.05).’ Perhaps the 'a' should be removed from this figure?
Discussion
Please provide information on how the nutrient composition of the feed changes during the development of the fish (endogenous, mixed and exogenous nutrition), including, if possible, the feed used in your experiment. At what stage of development are lipids or proteins the predominant components of the feed? This information may be useful to facilitate the discussion.
Although, as a non-native speaker, I am not qualified to judge the quality of the English in the paper, I feel that the article has serious flaws in English grammar as well as scientific style.
Author Response
Response to Reviewer 2 Comments
Point 1: Line 3: in ‘Endangered’ please use lower case.
Response 1: As suggested, we have changed 'Endangered' to lowercase(Line 3).
Point 2: Lines 18-22: Obviously, the study of intestinal tissues, digestive and antioxidant enzyme activities and malondialdehyde content is intended to assess the well-being of early juveniles of B. tsinlingensis in order to improve the efficiency of aquaculture of the species, rather than to ‘increase survival’ or ‘protect species resources and enrich the species' quantity’. Please rephrase.
Response 2: Thank you for your suggestion. But we believe that the ultimate goal of this article is to improve survival rates, protect species resources, and enrich species populations. Therefore, we have revised it to improve the aquaculture efficiency of this species, thereby increasing survival rates and pro-tecting species resources and enriching species numbers.
Point 3: Line 31: Please decipher the abbreviation TP.
Response 3: As suggested, We have explained the meaning of TP as total protein(Line 33).
Point 4: Line 61: Please provide full species name at the first mention as follows (indicating author) Brachymystax tsinlingensis Li, 1966.
Response 4: As suggested, We have provided the complete species name when first mentioned(Line 72).
Point 5: Line 100: Please provide a brief description of nutrient composition of the feed given to the fish at 45-73 DAH. Such data would facilitate discussion of the results obtained.
Response 5: As suggested, we have provided the nutrient content table for fish feeds(Lines 122-125 and 131-132).
Point 6: Line 102: Was the fish collected whole (or as questionably stated 'six tails were collected') for enzyme activity assays? You refer to 'intestinal enzymes' throughout the text; please provide information on the separation of intestinal enzymes or the dissection of the intestinal contents.
Response 6 : We collected samples from ten stages, six tail replicates per stage, and enzyme activity was assayed by weighing 0.1 to 0.2 samples from commissural fish, followed by mixing with frozen saline, high-speed grinder 2500 rpm, centrifuged for 10 minutes, and taking supernatant for assaying.
Point 7: Line 111: Confusingly, you refer to Elisa tests for lipase, amylase, trypsin, catalase, glutathione peroxidase and superoxide dismutase, as you obviously measured the enzyme activities but not their contents. Please specify the method applied and give the names of the kits used.
Response 7: As suggested, we have described the methods used and provided the names of the kits used(Lines 140-155).
Point 8: Line 143: It is recommended that the authors consider dividing the data in Table 1 into developmental stages, which would include the following categories: endogenous nutrition (7, 10, 11 days after hatching), mixed nutrition (13, 19 DAH) and exogenous nutrition (31, 33, 39, 45, 73 DAH).
Response 8: As suggested, We have divided the data in Table 1 into developmental stages (Lines 182-183).
Point 9: Line 149: in ‘The total protease concentration showed an overall upward trend …’. Replace ‘protease’ with ‘protein’.
Response 9: As suggested, We have replaced ‘protease’ with ‘protein’ (Line 188).
Point 10: Lines 169, 202: the figure captions ‘Trends in intestinal digestive enzymes of Total protein concentration’ and ‘Trends in intestinal antioxidant enzymes of Malondialdehyde content’ seem confusing and the word ‘trends’ is excessive; please rephrase.
Response 10: As suggested, We have rephrased the title in the picture(Lines 208, 246).
Point 11: Line 199: Figure 4B (catalase) has an ‘a’ over all the points, which means no difference. You write: ‘CAT activity decreased after the disappearance of yolk sac (19 DAH), but that increased after reaching the lowest peak at 39 DAH, and there was no significant difference in other groups (p > 0.05).’ Perhaps the 'a' should be removed from this figure?
Response 11: As suggested, We have removed 'a'.
Point 12: Please provide information on how the nutrient composition of the feed changes during the development of the fish (endogenous, mixed and exogenous nutrition), including, if possible, the feed used in your experiment. At what stage of development are lipids or proteins the predominant components of the feed? This information may be useful to facilitate the discussion.
Although, as a non-native speaker, I am not qualified to judge the quality of the English in the paper, I feel that the article has serious flaws in English grammar as well as scientific style.
Response 12: As suggested, We will provide feed for endogenous, mixed, and exogenous nutrients used during fish development.
We apologize for our negligence. We will strive to improve the deficiencies in English grammar and scientific style in the manuscript.
Reviewer 3 Report
Comments and Suggestions for Authors
Journal of Animals
This manuscript articulates a novel vision on of the endangered fish species. However, there are some comments raised in this study as follows:
- use a popular common name of the fish in the title, abstract , and introduction sections.
- L17: what grade in the globe?
- Please clarified how digestive system measurements can help researchers to find out artificial breeding attributes and population conservation of an endangered or wild fish?
- L31: TP, explain the acronyms fully once. Please double check this issue throughout the manuscript.
- L39: It is a big claim. How can it possible from the results? this is a limited study with limited samples and time. Re-write the statement.
- L62: I found that the fish species is native to rivers and lakes in Mongolia, Kazakhstan, wider Siberia (including Russian Far East), Northern China and Korea. Double check
- In the introduction section, the authors should clarify the reasons by giving relevant literature that Intestinal tissue, digestive enzyme and antioxidant capacity in in the early life stage of fish can provide insightful information to the artificial propagation and rearing (aquaculture operations) of endangered fish species.
- What was the diameter of tissue sections in this study?
- L116: What do you mean by 6 times? it means 6 samples for each treatment or repeating a method for one sample for 6 times?
- Elisa tests were performed for serum or intestine and why?
- Why the authors did not measure the antioxidant enzymes activity in the serum instead on intestine?
- Which part of the gastrointestinal was used ? and why?
- In terms of intestinal tissue structure, which region (fore, mid, etc) was used? and why?
- All the data were normal to perform ANOVA? If yes how can the authors approve this issue?
- Why the authors did not provide the growth indices? It is interesting to know the growth rate of this species.
- what is the difference between Figs. 2 and 3? similar to Figs. 4 and 5. It is NOT necessary to repeat the results in different format or graphs, generally.
- The authors must provide proximate composition of the diets.
- There is no need to separate the discussion sections by subtitles. In research-original papers, it is not common to classify the discussion section by subtitles. Because each paragraph in this section can show the discussion of each physiological pathway(s).
- L 307: any recommendation for the future research ?
Author Response
Response to Reviewer 3 Comments
Point 1: use a popular common name of the fish in the title, abstract , and introduction sections.
Response 1: As suggested, We have used the popular generic name of fish in the title, abstract, and introduction sections.
Point 2: Line 17: what grade in the globe?
Response 2: We apologize for our negligence. We found no articles where Brachymystax tsinlingensis belong globally. But we're confident that the wild population of Brachymystax tsinlingensis is a Grade â…¡ protected species in China.
Point 3: Please clarified how digestive system measurements can help researchers to find out artificial breeding attributes and population conservation of an endangered or wild fish?
Response 3: We thank the reviewer for the valuable comment. In order to solve the problem of low survival rate during the overwintering period of artificial propagation of Brachymystax tsinlingensis, it is necessary to analyze its intestinal tissue structure, whose villus height, width and thickness of muscle layer can show the degree of food digestion of the fry, and ultimately achieve the purpose of improving the survival rate, protecting the species' resources and enriching the species' population.
Point 4: Line 31: TP, explain the acronyms fully once. Please double check this issue throughout the manuscript.
Response 4: We apologize for our negligence. We have double-checked for complete acronyms throughout the manuscript.
Point 5: Line 39: It is a big claim. How can it possible from the results? this is a limited study with limited samples and time. Re-write the statement.
Response 5: We thank the reviewer for the valuable comment. This experiment was completed under the artificial propagation of wild Brachymystax tsinlingensis, and the authors themselves participated in the whole process from parental selection to spawning, incubation and cultivation. So the sample collection stage was 7-73 DAH, and 6 tails were collected at each stage of the experiment, which meets the requirements of experimental data analysis.
Point 6: Line 62: I found that the fish species is native to rivers and lakes in Mongolia, Kazakhstan, wider Siberia (including Russian Far East), Northern China and Korea. Double check
Response 6: Thank you for your suggestion. The Brachymystax tsinlingensis we studied it is endemic to China, as a subspecies of Brachymystax previously (Brachymystax tsinlingensis), and only found in parts of the Taibaishan Moutains segment of the Qinling Mountains, especially in the Heihe, Shitouhe, Xushui and Taibai Rivers (Li, 1966), as a residual species migration from the north southward during glacial period(Xiong et al., 2019; Xiong et al., 2023).
Point 7: In the introduction section, the authors should clarify the reasons by giving relevant literature that Intestinal tissue, digestive enzyme and antioxidant capacity in in the early life stage of fish can provide insightful information to the artificial propagation and rearing (aquaculture operations) of endangered fish species.
Response 7: Thank you for your suggestion. We have provided relevant literature in the introduction section to elucidate that gut organization, digestive enzymes, and antioxidant capacity in early stages of fish life can support captive breeding and rearing of endangered fish species.
Point 8: What was the diameter of tissue sections in this study?
Response 8: Thank you for your suggestion. We have supplemented the diameter data of the tissue sections (Line 135).
Point 9: Line 116: What do you mean by 6 times? it means 6 samples for each treatment or repeating a method for one sample for 6 times?
Response 9: Thank you for your suggestion. The six times we propose refer to six samples
Point 10: Elisa tests were performed for serum or intestine and why?
Response 10: Thank you for pointing out those questions for us. The Elisa test is conducted on the intestine to analyze the digestive ability of fish fry during the early development stage of Brachymystax tsinlingensis. The aim is to improve digestion rate, reduce mortality rate, and ultimately increase species abundance.
Point 11: Why the authors did not measure the antioxidant enzymes activity in the serum instead on intestine?
Response 11: We thank the reviewer for the valuable comment. The activity of antioxidant enzymes in serum is used to test the immune ability of fish fry, while the detection of antioxidant enzyme activity in the intestine, which is used to test the strength of oxidative stress damage caused by feeding feed to fish fry, in order to regulate the subsequent feeding of feed to fry.
Point 12: Which part of the gastrointestinal was used ? and why?
Response 12: We thank the reviewer for the valuable comment. Using the entire intestinal tissue, as Brachymystax tsinlingensis is a cold water fish with slow growth, it is difficult to complete the testing requirements by taking some intestinal tissue.
Point 13: In terms of intestinal tissue structure, which region (fore, mid, etc) was used? and why?
Response 13: We thank the reviewer for the valuable comment. In terms of intestinal tissue structure, all intestinal tissue is used for embedding, as fish fry are too small and can easily lose tissue after dehydration. Slices are displayed in cross-section.
Point 14: All the data were normal to perform ANOVA? If yes how can the authors approve this issue?
Response 14: We thank the reviewer for the valuable comment. All data is normal and can be analyzed for variance.
Point 15: Why the authors did not provide the growth indices? It is interesting to know the growth rate of this species.
Response 15: We thank the reviewer for the valuable comment. The Brachymystax tsinlingensis is a cold water fish with slow growth, and its growth index requires more time and sample support. Of course, our research group will also consider related studies in the future.
Point 16: what is the difference between Figs. 2 and 3? similar to Figs. 4 and 5. It is NOT necessary to repeat the results in different format or graphs, generally.
Response 16: We thank the reviewer for the valuable comment. We believe that Figure 2 clearly expresses a line graph of the changes in enzyme activity in each period, while Figure 3 shows the percentage of various enzymes by considering each indicator in the same period as a whole.
Point 17: The authors must provide proximate composition of the diets.
Response 17: As suggested, We have provided an approximate composition of the diet (Lines 122-125 and 131-132).
Point 18: There is no need to separate the discussion sections by subtitles. In research-original papers, it is not common to classify the discussion section by subtitles. Because each paragraph in this section can show the discussion of each physiological pathway(s).
Response 18: Thank you for your suggestion. We have removed the subheadings from the discussion section
Point 19: Line 307: any recommendation for the future research?
Response 19: Thank you for your suggestion. We can conduct similar experiments in the future to find the most suitable feed to reduce oxidative stress damage in Brachymystax tsinlingensis fry.

Reviewer 4 Report
Comments and Suggestions for Authors
Manuscript ID: animals-3180100
Type: Article
Title: Intestinal tissue, digestive enzyme and antioxidant enzyme activities in the early development stage of Endangered Brachymystax tsinlingensis
Review observations
Line 31: Define TP.
Line 31: Please clarify this paragraph. Indicate the activities of Lipase (xxx U/mg protein), Trypsin (xxx U/mg protein), etc. found.
Line 33: (0.57±0.11 U/mg and 3.35±0.94 U/mg, respectively).
Line 34-35: (1.32±0.41 U/mg) and 73 DAH (1.29±0.43 nmoL/mg, respectively).
Line 36: (126.58±20.13 U/mg and 6.47±1.86 U/mg, respectively).
Line 53: Indicate which intestinal enzymes.
Line 56-57: Define digestive capacity.
Line 63: If B. tsinlingensis is a carnivorous fish, are stomach enzymes important for digestive capacity? please include this information in text. Explain when stomach digestive capacity is important in B. tsinlingensis.
Line 73: Indicate which activities.
Line 74: Indicate which digestive enzymes and antioxidant enzymes are.
Line 76: Indicate in the text why the study of stomach and stomach digestive enzyme activity was not included.
Line 112-115: For enzyme activity assays, include the substrate used, assay temperature, and enzyme unit definitions.
Line 167: Fig 2. In Y axes, g or mg of what.
Line 200: Fig 2. In Y axes, g or mg of what.
Line 235: Please discuss total enzyme activity values ​​(U/larvae and/or U/g of larvae).

Author Response
Response to Reviewer 4 Comments
Point 1: Line 31: Define TP.
Response 1: We apologize for our negligence. We have added the definition of TP (Line 33).
Point 2: Line 31: Please clarify this paragraph. Indicate the activities of Lipase (xxx U/mg protein), Trypsin (xxx U/mg protein), etc. found.
Response 2: We apologize for our negligence. We have clarified the activities of total protein, lipase, and trypsin (Line 32).
Point 3: Line 33: (0.57±0.11 U/mg and 3.35±0.94 U/mg, respectively).
Response 3: We apologize for our negligence. We have completed the modifications (Line 34).
Point 4: Lines 34-35: (1.32±0.41 U/mg) and 73 DAH (1.29±0.43 nmoL/mg, respectively).
Response 4: We apologize for our negligence. We have completed the modifications (Lines 35-36).
Point 5: Line 36: (126.58±20.13 U/mg and 6.47±1.86 U/mg, respectively).
Response 5: We apologize for our negligence. We have completed the modifications (Line 37).
Point 6: Line 53: Indicate which intestinal enzymes.
Response 6: We apologize for our negligence. Intestinal enzymes include amylase, protein, trypsin, chymotrypsin and lipase (Lines 56-57).
Point 7: Lines 56-57: Define digestive capacity.
Response 7: We apologize for our negligence. The digestive capacity from the article Functional changes in digestive enzyme activities of meagre(Argyrosomus regius; Asso, 1801) during early ontogeny.
Point 8: Line 63: If B. tsinlingensis is a carnivorous fish, are stomach enzymes important for digestive capacity? please include this information in text. Explain when stomach digestive capacity is important in B. tsinlingensis.
Response 8: We thank the reviewers for their valuable comments. In the future, we consider adding gastric structural enzyme activity assays to our experiments to fill the gaps in our current study.
Point 9: Line 73: Indicate which activities.
Response 9: The activity refers to the study of intestinal tissue, digestive enzymes (total protein, lipase, trypsin, and amylase), and antioxidant enzymes (catalase, glutathione peroxidase, and superoxide dismutase) in the early developmental stages of Brachymystax tsinlingensis.
Point 10: Line 74: Indicate which digestive enzymes and antioxidant enzymes are.
Response 10: The digestive enzymes were total protein, lipase, trypsin and amylase. The antioxidant enzymes were catalase, glutathione peroxidase and superoxide dismutase.
Point 11: Line 76: Indicate in the text why the study of stomach and stomach digestive enzyme activity was not included.
Response 11: We thank the reviewers for their valuable comments. In the future, we consider adding gastric structural enzyme activity assays to our experiments to fill the gaps in our current study.
Point 12: Line 112-115: For enzyme activity assays, include the substrate used, assay temperature, and enzyme unit definitions.
Response 12: Thank you for your suggestion. We have supplemented the specific enzyme activity assay (2.4Elisa test) with the substrates, assay temperatures and enzyme unit definitions used for the experiments, which were performed in accordance with the kit requirements.
Point 13: Line 167: Fig 2. In Y axes, g or mg of what.
Response 13: Thank you for your suggestion. We have standardized the enzyme activity units to mg. And the mg on the Y-axis is the unit specified in Nanjing Jianjian Biological Company's enzyme activity test kit.
Point 14: Line 200: Fig 2. In Y axes, g or mg of what.
Response 14: Thank you for your suggestion. We have standardized the enzyme activity units to mg. The mg on the Y-axis is the unit specified in Nanjing Jianjian Biological Company's enzyme activity test kit.
Point 15: Line 235: Please discuss total enzyme activity values (U/larvae and/or U/g of larvae).
Response 15: Thank you for your suggestion. We believe that the value of total enzyme activity is analyzed from the stage and overall respectively. Firstly, from the stage, the trend of enzyme activity can visualize the body's digestion and absorption of food. Secondly, from the overall, the trend of total enzyme activity can be used to make appropriate bait choices for its early developmental stage, so as to enhance the digestive ability and promote the growth and development of the fish.

Round 2
Reviewer 2 Report
Comments and Suggestions for Authors
The revised version of the manuscript “Intestinal tissue, digestive enzyme and antioxidant enzyme activities in the early development stage of endangered Brachymystax tsinlingensis” by Rongqun Song et al. is significantly improved by author revisions. However, some of the revisions are formal and tend to confuse the collocations.
Lines 32-34, 50, 145, 208. You use ‘activity’ instead of ‘content’ or ‘level’ when discussing total protein. It is not a digestive enzyme, as you wrote; please rephrase.
Lines 56-57, 95-96, 150, 208-209, 214, 241-242, 246-247. Enzyme names should not be capitalised;
Table 1 – seemed unnecessary. Better to give this information in a line without formatting in the table and not in bold.
Line 144. You write ‘malondialdehyde as a marker of antioxidant substances’, whereas MDA is a product of fatty acid oxidation and is used as a marker of oxidative stress.
Figure 2. The sector B is given as ‘lipase activity’ in the figure caption, while ‘trypsin’ is given in a figure (as a legend and axis name); please, revise.
Line 292. What is ‘CP level’? it’s not mentioned above.
Line 323. Typo in ‘So, the oxidative stress damage of B. tsinlingensis ware gradually serious’
Line 349. The phrase ‘We can conduct similar experiments in the future to find the most suitable feed to reduce oxidative stress damage in B. tsinlingensis fry’ needs to be revised.
I strongly recommend that the entire text be revised for consistency. Although the scientific content of the MS and the data discussion are satisfactory, the text needs to be carefully checked before publication.
Author Response
Response to Reviewer 2 Comments
Point 1: Lines 32-34, 50, 145, 208. You use ‘activity’ instead of ‘content’ or ‘level’ when discussing total protein. It is not a digestive enzyme, as you wrote; please rephrase.
Response 1: Thank you for your suggestion. We have standardized the wording to total protein content (Lines 32-37, 95-97, 146-147 and 210-211).
Point 2: Lines 56-57, 95-96, 150, 208-209, 214, 241-242, 246-247. Enzyme names should not be capitalised;
Response 2: Thank you for your suggestion. We have lower-cased the enzyme name (Lines 57-58, 95-97, 150, 210-211, 215-216, 242-243 and 247-249).
Point 3: Table 1 – seemed unnecessary. Better to give this information in a line without formatting in the table and not in bold.
Response 3: Thank you for your suggestion. We have converted Table 1 into a textual representation (Lines 125-128).
Point 4: Line 144. You write ‘malondialdehyde as a marker of antioxidant substances’, whereas MDA is a product of fatty acid oxidation and is used as a marker of oxidative stress.
Response 4: As suggested, we have changed ‘malondialdehyde as a marker of antioxidant substances’ to ‘malondialdehyde is a marker of oxidative stress’ (Line 145).
Point 5: Figure 2. The sector B is given as ‘lipase activity’ in the figure caption, while ‘trypsin’ is given in a figure (as a legend and axis name); please, revise.
Response 5: We apologize for our negligence. We have checked and corrected the pictures and icons (Lines 208 and 240).
Point 6: Line 292. What is ‘CP level’? it’s not mentioned above.
Response 6: The ‘CP level’ indicates ‘Larvae were fed to apparent satiation four times per day with non-enriched Artemia nauplii (O.S.I. PRO 80™, Ocean Star International, Inc. USA) from mouth opening (2 dph) until 10 dph. From 7 dph onwards, zooplankton collected from nearby ponds, which consisted mainly of copepods (Cyclopoida), were also added to fish rearing tanks. After 10 dph, only zooplankton was given’(Pradhan et al., 2013) and ‘Four experimental diets differing in CP level were formulated: 30%, 36%, 42% and 48%’(Silva et al., 2019).
Pradhan, P.; Jena, J.; Mitra, G.; Sood, N.; Gisbert, E. Ontogeny of the digestive enzymes in butter catfish Ompok bimaculatus (Bloch) larvae. Aquaculture, 2013, 372-375: 62-69.
Silva, W.; Costa, L.; Lopezolmeda, J.; Costa, N.; Santos, W.; Ribeiro, P.; Luz, R. Gene expression, enzyme activity and performance of Nile tilapia larvae fed with diets of different CP levels. Animal: an international journal of animal bioscience 2019, 13, 1376-1384.
Point 7: Line 323. Typo in ‘So, the oxidative stress damage of B. tsinlingensis ware gradually serious’
Response 7: As suggested, we have changed ‘So’ to ‘Therefore’ (Line 323).
Point 8: Line 349. The phrase needs to be revised.
I strongly recommend that the entire text be revised for consistency. Although the scientific content of the MS and the data discussion are satisfactory, the text needs to be carefully checked before publication.
Response 8: As suggested, we have changed ‘We can conduct similar experiments in the future to find the most suitable feed to reduce oxidative stress damage in B. tsinlingensis fry’ to ‘It is hoped that appropriate bait feeding can be found to reduce oxidative stress damage in the subsequent captive breeding of B. tsinlingensis’ (Lines 349-351).
We tried our best to improve the manuscriptand made some changes in the manuscript. These changes will not influence the content and framework of the paper. We appreciate for Editors/Reviewers 'warm workearnestly, and hope that the correction will meetwith approval.
Reviewer 3 Report
Comments and Suggestions for Authors
No further comments.
Author Response
We would like to thank you for your professional review work, constructive comments, and valuable suggestions on our manuscript, which would help to improve the quality of our manuscript.